# Influence of Residual Stress on Fatigue Weak Areas and Simulation Analysis on Fatigue Properties Based on Continuous Performance of FSW Joints

**Guoqin Sun [1],\*, Xinhai Wei [1], Jiangpei Niu [2], Deguang Shang [1] and Shujun Chen [1]**

[1]  College of Mechanical Engineering and Applied Electronics Technology, Beijing University of Technology, Beijing 100124, China; 18810816630@163.com (X.W.); shangdg@bjut.edu.cn (D.S.); sjchen@bjut.edu.cn (S.C.)

[2]  Production R&D center, CRRC Tangshan Co. Ltd., No.3 Changqian Rd, Tangshan 064000, China; sjc-niujiangpei@tangche.com

\*  Correspondence: sguoq@bjut.edu.cn; Tel.: +86-010-6739-6888

**Abstract:** The fatigue weak area of aluminum alloy for a friction stir-welded joint is investigated based on the hardness profile, the residual stress measurement and the simulation analysis of fatigue property. The maximum residual stresses appeared at the heat-affected zone of the joint in the fatigue damage process, which was consistent with the fracture location of the fatigue specimen. The fatigue joint model of continuous performance is established ignoring the original negative residual stress; considering that it will be relaxed soon when the joint is under tension-tension cyclic loading. The fatigue parameters of joint model is based on the static mechanical properties of the joint that obtained from the micro-tensile tests and four-point correlation method. The predicted results for the fatigue weak locations and fatigue lives based on the continuous performance joint model are closer to the fatigue experimental results by comparison with the simulation results of the partitioned performance joint model.

**Keywords:** friction stir welding; residual stress; weak area; finite element simulation; life prediction

## 1. Introduction

Friction stir welding (FSW) of aluminum alloy has been widely used in the automotive, aerospace and ship industries. The anti-fatigue design is worthy of attention for the load-carrying FSW components. The influencing factors on the failure location of the joints are needed to be focused on. The hardness distribution of the friction stir-welded joints is related to the materials and welding process. It is considered that it has a relation to the fatigue fracture position of alloy joints [1–6]. But some studies showed that fatigue cracks were independent from hardness distributions in the weld seams and the fatigue cracks initiated at the inhomogeneous microstructure [7]. Fatigue fracture of the large plate happened at the lowest hardness location in the heat affect zone (HAZ) [8]. But the tensile strength of the FSW joint has a linear relationship with the weld nugget hardness [9]. The fatigue properties of aluminum alloy FSW joints are also affected by the residual stress [2–4]. The residual stresses of aluminum alloy welded joints are dependent on the welding parameters and their distribution has a typical "M" profile [10–13]. The locations of the maximum residual tensile stress are different for different alloy joints. The compressive residual stress delays the crack growth and increases the fatigue life of the welded joint and the tensile residual stress accelerates the propagation of the crack [14–17]. Fratini et al. [4] found that the residual stress had an influence on the crack propagation of base metal area and had no obvious effect on the welding area. The effect of residual stress on the joint would weaken with the increase of fatigue cycle or the crack length [18,19].

Finite element numerical simulation has been applied on the evaluation of mechanical properties for FSW joints. In addition, the residual stress can be preloaded into the joint model as a prestress if it is non-negligible [20]. Rao and Simar [21,22] established the finite element model of the FSW joint through the partition method to obtain the tensile properties. The friction stir welding joints were divided into different regions to simulate the mechanical properties of the weak regions [23,24].

The simulation method with the partitioned performance joint model has the stress concentration at the regional boundary positions because of the discontinuity of material properties. The continuous performance joint model does not have the effect of the area partition and obtains more accurate stress–strain data. Therefore, the fatigue finite element model of continuous performance for the FSW joint is established to simulate the weak areas and the stress–strain response. The life prediction is proceeded based on the simulation results and a comparison between the continuous performance joint model and the partitioned performance joint model is carried out. Also, the relationship between the fatigue weak areas of 2219-T6 aluminum alloy friction stir-welded joints and mechanical properties is analyzed with experimental.

## 2. Experimental

### 2.1. Materials and Specimens

The test specimens is cut from 2219-T6 aluminum alloy FSW butt plate of 800 mm × 300 mm × 6 mm size. The chemical compositions and mechanical properties of base metal are given in Tables 1 and 2. The alloy plates were welded perpendicular to the rolling direction with a rotation speed of 800 rpm and an advancing speed of 180 mm/min. The FSW tool consists of a shoulder with a diameter of 18 mm and a tool pin of 5.7 mm in length and 6 mm in diameter. The tool axis is tilted by 2° with respect to the vertical axis of the plate surface. The welding was finished in the China Academy of Launch Vehicle Technology. The fatigue samples were wire cut from the plate, which its axial direction was perpendicular to the weld line. The surfaces and sides of the specimens were ground with silicon carbide sandpaper from 150 to 2000 grit and the surfaces were polished with diamond polishing pastes from 4000 to 10,000 grit. The thickness of the fatigue specimen is 6 mm and the dimensions is shown in Figure 1.

**Table 1.** Chemical composition of 2219 aluminum alloy.

| Chemical Composition (wt%) | | | | | | | | |
|---|---|---|---|---|---|---|---|---|
| Cu | Mn | Fe | Zn | Si | Zr | Ti | V | Al |
| 6.48 | 0.32 | 0.23 | 0.04 | 0.49 | 0.2 | 0.06 | 0.08 | balance |

**Table 2.** Mechanical properties of 2219-T6 aluminum alloy.

| Material | Elasticity Modulus (GPa) | Ultimate Strength $\sigma_u$ (MPa) | Yield Strength $\sigma_{p0.2}$ (MPa) |
|---|---|---|---|
| 2219-T6 | 72 | 416 | 315 |

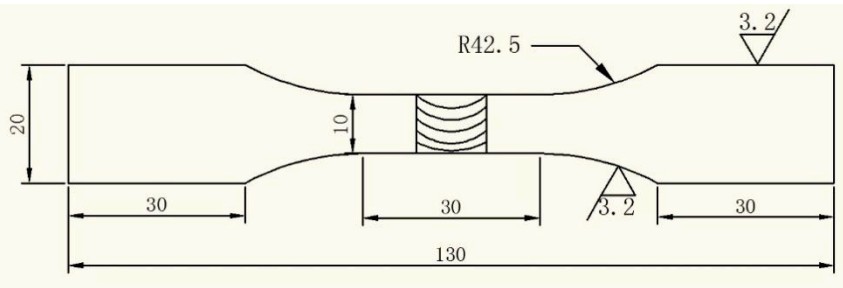

**Figure 1.** Shape and size of fatigue specimen.

*2.2. Metallographic Morphology*

The cross-sectional metallographic morphology of the specimen was observed with the optical microscope (Shanghai optical instrument factory, Shanghai, China) after the specimen was etched with keller's reagent consisting of 2.5 mL $HNO_3$, 1.5 mL HCL, 1 mL HF and 95 mL $H_2O$ for about 15 s. According to the metallographic morphology of the welded joint in Figure 2 and the sizes of the grain, the welded joint is divided into different regions: the weld nugget zone (WNZ), the thermo-mechanically affected zone (TMAZ), the HAZ and the base material (BM). The HAZ is further divided into high-hardness heat affect zone (HHAZ) and low-hardness heat affect zone (LHAZ) based on the hardness distribution and the phase sizes. The phase in the HHAZ has the similar size as that in the BM, while the hardness in HHAZ is lower than that of BM. In addition, according to the rotation direction of the welding tool, the joint is divided into the advancing side (AS) and the retreating side (RS). The dimensions of the different zones that marked in the hardness profiles can be obtained by combining the metallographic morphology, phase size and the hardness profile.

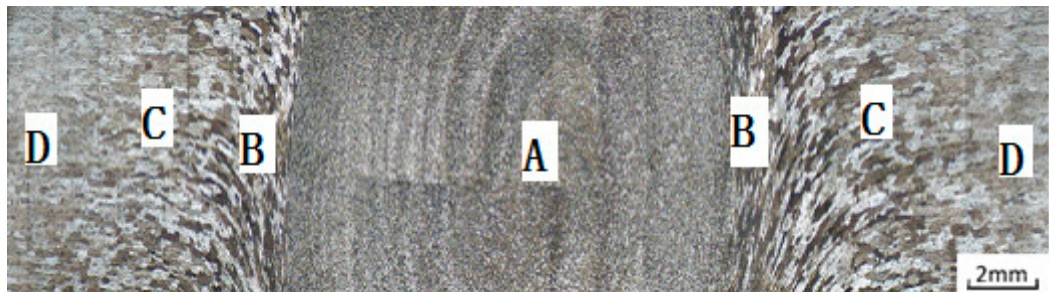

**Figure 2.** Metallograph of the aluminum alloy FSW joints: (**A**) WNZ, (**B**) TMAZ, (**C**) HAZ, (**D**) BM.

*2.3. Hardness Measurements*

The microhardness distributions were measured on the upper and the lower surfaces of the welded joint with a load of 1.95 N for 15 s with a SCTMC DHV-1000Z Vickers hardness tester (Shangcai tester machine Co, Ltd, Shanghai, China). The measurement spacing between each two adjacent points was 0.5 mm.

*2.4. Fatigue Experiments*

All fatigue tests were carried out using MTS858 hydraulic servo system (MTS Systems Corporation, Minneapolis, MN, USA) under the axial stress range from 216 to 261 MPa with a stress ratio of 0.1 and a frequency of 10 Hz until the specimens fractured. The fatigue loading parameters and total fatigue lives are available from Ref. [25]. The curve of cyclic stress range versus fatigue lives for the FSW specimens is shown in Figure 3.

*2.5. Residual Stress Measurements*

The residual stresses of the original and the fatigue-damaged specimens were measured by D8 Discover X-ray diffraction meter (Bruker Corporation, Karlsruhe, Germany). The specimen was electrochemically polished to remove the effect of mechanical polishing before the residual stress measurement. The original residual stress distributions of the upper and the lower surfaces for the joint were measured before fatigue test. Then, the cyclic stress range of 216 MPa with a stress ratio of 0.1 was loaded on the joint and the fatigue test was stopped when the fatigue life was 30,000 cycles. Then, the surface residual stresses of the fatigue-damaged specimen were measured again to analyze the effect of the residual stress distribution in the fatigue process on fatigue damage.

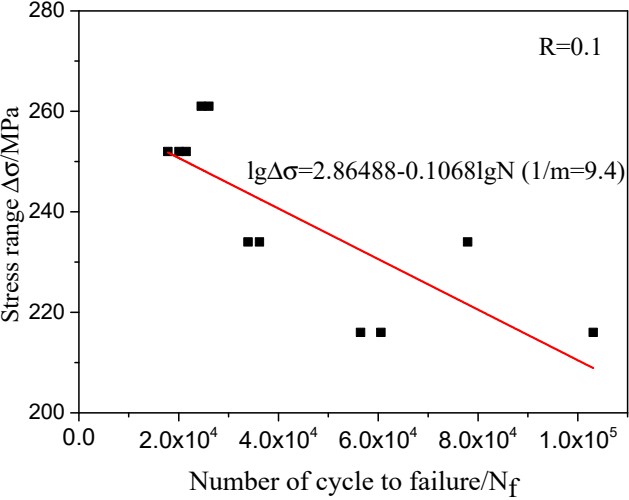

**Figure 3.** S-N curve of FSW specimens.

## 3. Experimental Results

### 3.1. Hardness Distribution

The upper and the lower surfaces hardness profiles of the welded joint are shown in Figure 4. The hardness distributions of the joint upper and lower surfaces present the approximate "W" shape. The hardness in the welded region is obviously lower than the base material. The minimum hardness values of the upper and the lower surfaces are both in the LHAZ according to the hardness distribution of the joints, which is not in coincidence with the failure position of HHAZ for most fatigue specimens. Hardness has a relation with the material strength but the fatigue weak area of the joint is not necessarily corresponding to the location of minimum hardness [7]. It has a relation with the variation of the hardness gradient, which corresponds to the variation of the mechanical property and heterogeneous microstructure.

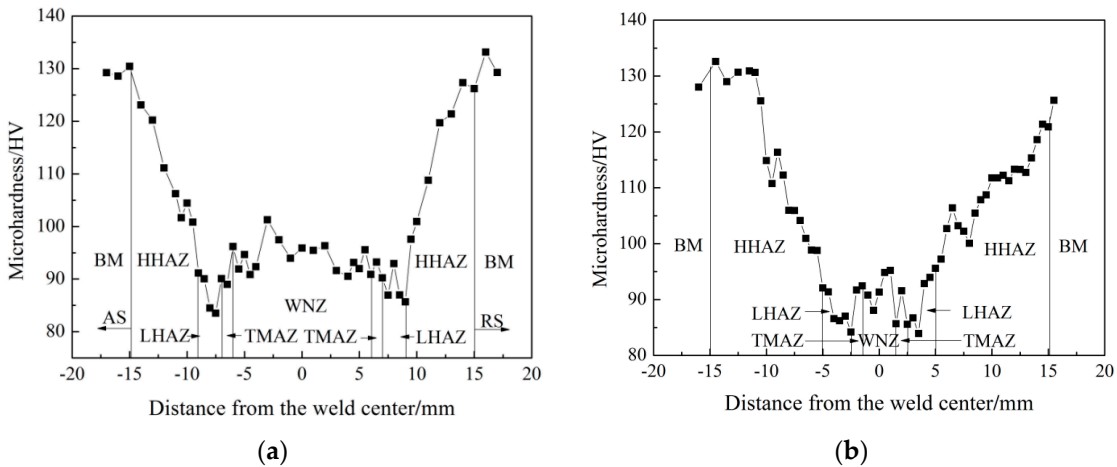

**Figure 4.** Hardness distributions of (**a**) upper and (**b**) lower surfaces in the joint.

### 3.2. Fatigue Experimental Results

The failure locations in the joints were observed with the optical microscope after the specimens fractured. First, the locations of the fatigue crack sources are needed to be found and observed. Then, the sides or the surfaces of the sample near the location of fatigue crack sources are etched with Keller's reagent. The failure locations can be affirmed through the metallographic morphologies of the samples.

The statistics of failure locations for 11 specimens are listed as follow: 10 specimens fractured in HHAZ, 1 specimen broke in WNZ.

### 3.3. Residual Stress Distribution

The residual stress distributions are shown in Figures 5 and 6. The transverse residual stresses are the stresses perpendicular to the weld and the longitudinal residual stresses are the stresses parallel to the weld direction. The center of the WNZ was taken as the zero coordinate and the test points were selected on both sides along the axial direction of the specimen. The measured residual stresses before fatigue test are called as the original residual stresses of the specimen in this paper. The residual stresses of the fatigue-damaged specimens were measured after the fatigue cycle reached 30,000 cycles under the cyclic stress range of 216 MPa with a stress ratio of 0.1.

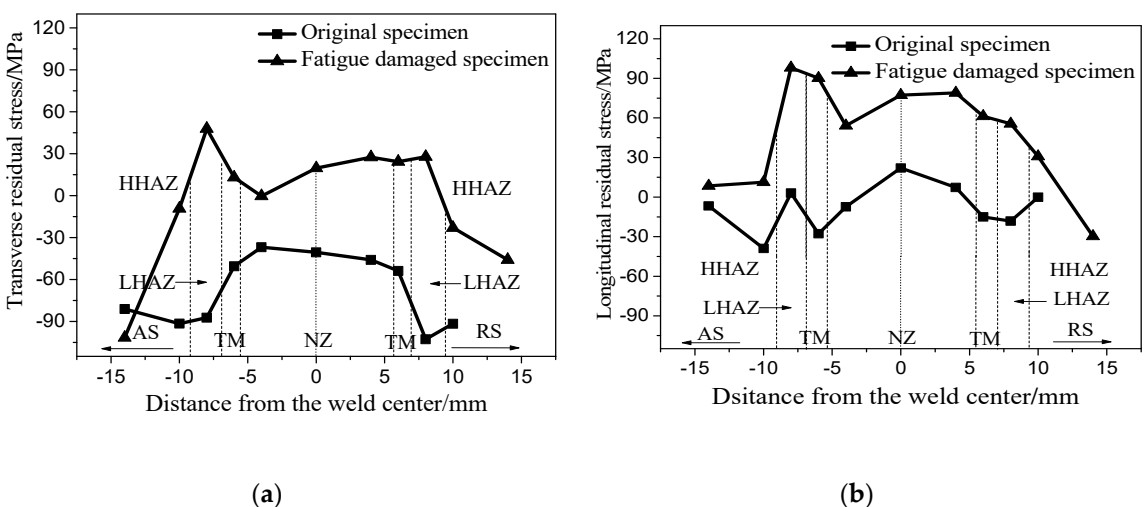

(**a**)             (**b**)

**Figure 5.** (**a**) Transverse and (**b**) longitudinal residual stress distributions of upper surface in the joint.

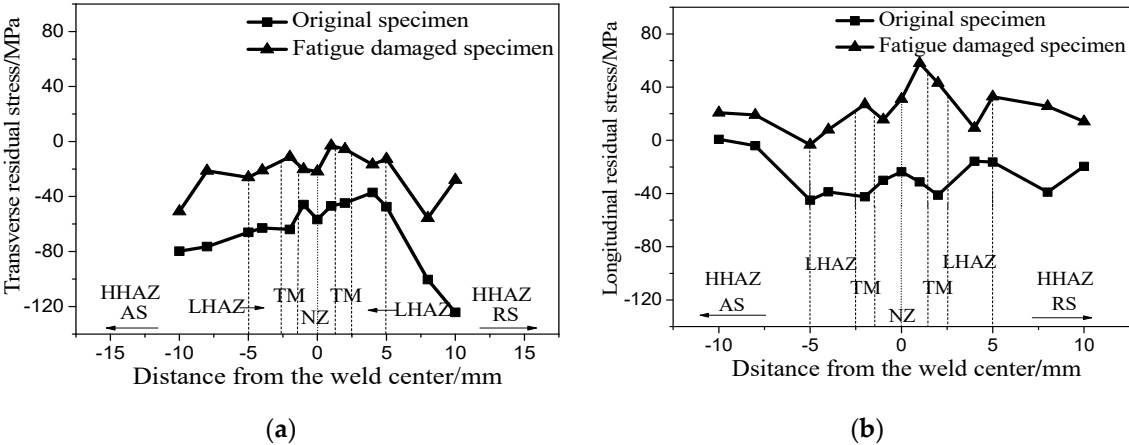

(**a**)             (**b**)

**Figure 6.** (**a**) Transverse and (**b**) longitudinal residual stress distributions of lower surface in the joint.

Figure 5 shows the variation of the transverse and longitudinal residual stress distribution curves in the upper surface of the welded joint. It can be seen that the maximums of the transverse and longitudinal residual stresses in the original welded joint appear in the WNZ and the values are basically negative or near zero. The original negative residual stresses in the welded joint is beneficial to the fatigue life. However, the original compress residual stress will be relaxed and the positive stress will appear soon once the tension-tension stress is loaded. The positive transverse and longitudinal residual stresses arose in the fatigue process and the maximum residual stress appeared at the LHAZ during the cyclic loading.

Figure 6 shows the variation of the transverse and longitudinal residual stress distribution curves in the lower surface of the welded joint. The residual stresses is negative in the original joint. The maximum stresses appeared near the boundary of NZ and TMAZ in the fatigue process and the values were far lower than the residual stresses in the upper surface. The residual stress of the upper surface in the joint that produced in the fatigue process should have more effect on the fatigue damage.

## 4. Fatigue Numerical Simulation

### 4.1. Fatigue Parameters of the Welded Joint

The fatigue finite element numerical analysis is further proceeded in order to obtain the stress–strain responses and evaluate the weak area of the joint with the simulation method. Since the original residual stress was negative and would be relaxed once the tension-tension stress was loaded. It did not considered to be added on the joint as a prestress in the numerical analysis under the tension-tension cyclic loading.

The fatigue parameters are different in different zones since the materials of different zones in the joint have different microstructures and mechanical properties. They can be used in some fatigue life prediction models to calculate fatigue lives or as the intermediate variables to calculate other cyclic strength parameters. The fatigue parameters of different positions in the welded joint were obtained by four-point correlation method proposed by Manson [26].

The four-point correlation method is based on the elastic and plastic strain-life lines. The respective two points on elastic and plastic lines are definite. On the elastic line, a point is at 1/4 cycle with a strain range of 2.5 $\sigma_f/E$, where $\sigma_f$ is the fracture strength, $E$ is the elastic modulus. The other point is at $10^5$ cycles with a strain range of 0.9 $\sigma_u/E$, where $\sigma_u$ is the ultimate tensile strength. On the plastic line, a point locates at 10 cycles with a strain range of $1/4D^{3/4}$, where $D$ is the logarithmic ductility of the material. Another point is at $10^4$ cycles with a strain range of $\frac{0.0132-\Delta\varepsilon_e^*}{1.91}$, $\Delta\varepsilon_e^*$ indicates the value of elastic strain range at $10^4$ cycles on the elastic line, as shown in Figure 7.

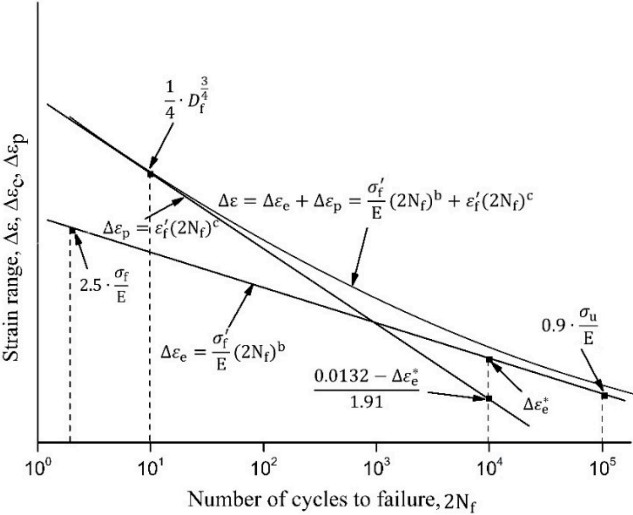

**Figure 7.** Four-point correlation method by Manson.

The static mechanical properties of different positions in the joint were obtained from the micro-tensile tests. Microspecimens from different regions were cut parallel to the weld direction of welded joints. The static mechanical properties of each zone were measured by using Instron 5948 Microtester (Instron Corporation, Norwood, MA, USA). The calculated fatigue parameters of different positions in the welded joint are listed in Table 3.

**Table 3.** Fatigue parameters of the welded joint.

| Cutting Cites of the Micro Specimens X (mm) | Elastic Modulus $E$ (GPa) | Fatigue Strength Coefficient $\sigma'_f$ (MPa) | Fatigue Strength Exponent $b$ | Fatigue Ductility Exponent $c$ | Fatigue Ductility Coefficient $\varepsilon'_f$ | Cyclic Strength Coefficient $K'$ (MPa) | Cyclic Strain Hardening Exponent $n'$ |
|---|---|---|---|---|---|---|---|
| −11 | 64 | 364 | −0.078 | −0.33 | 0.041 | 781 | 0.24 |
| −7 | 53 | 275 | −0.079 | −0.31 | 0.037 | 635 | 0.25 |
| −5 | 58 | 324 | −0.077 | −0.37 | 0.062 | 578 | 0.21 |
| 0 | 59 | 334 | −0.078 | −0.37 | 0.059 | 610 | 0.21 |
| 6 | 61 | 313 | −0.076 | −0.35 | 0.055 | 581 | 0.21 |
| 8 | 62 | 263 | −0.058 | −0.33 | 0.043 | 458 | 0.18 |
| 11 | 63 | 328 | −0.076 | −0.34 | 0.047 | 656 | 0.23 |

### 4.2. Continuous Performance Joint Model

The continuous performance joint model is established by inputting different elastic moduli and fatigue stress–strain data at different areas of the joint with ABAQUS software. The WNZ center of the joint is the zero coordinate of the X axis in the model. The different coordinates correspond to the different locations of the joint along the axial loading direction. The elastic moduli of different positions with the coordinate X that listed in Table 3 were obtained from the micro tension tests. The elastic moduli of the other positions in the joint can be obtained by interpolation according to the data in Table 3. The mechanical properties of the joint also vary with change of the coordinate. The materials in different zones of the joint have different stress–strain responses. The cyclic stress–strain at different locations of the joint were obtained using Ramberg–Osgood equation and applied to the finite element simulation. The Ramberg–Osgood equation is shown below.

$$\varepsilon_a = \frac{\sigma_a}{E} + \left(\frac{\sigma_a}{K'}\right)^{\frac{1}{n'}} \tag{1}$$

where $\varepsilon_a$ is the strain amplitude, $\sigma_a$ is the stress amplitude, $K'$ is cyclic strength coefficient, $n'$ is the cyclic strain hardening exponent. They can be obtained from the following equations.

$$K' = \frac{\sigma'_f}{\varepsilon'^{n'}_f} \tag{2}$$

$$n' = b/c \tag{3}$$

The cyclic yield strength was measured by the stress value of 0.2% plastic strain in the cyclic stress–strain curve. The nonlinear kinematic hardening model of the material attribute was adopted for the joint material.

User subroutine USDFLD in the ABAQUS software can be used to describe material properties. The material properties is defined as a function of field variables and the variables can be solved with USDFLD subroutine. The variation of elastic modulus, yield stress and plastic strain with the coordinate X were computed with the interpolation method using the known data of the specific coordinate through the subroutine.

The joint model is constrained at one end and loaded the cyclic stress range of 216 MPa with a stress ratio of 0.1 at another end of the joint in the X-axial direction. The loaded cyclic stress is schematically drawn in Figure 8. The hexahedron mesh with eight nodes element type of C3D8R is selected to solve the structure. The size of each element is approximately 1.3 mm × 0.8 mm × 1.2 mm. 3600 elements and 4758 nodes are obtained to simulate the FSW joints, as shown in Figure 9. The model does not include the clamping part and the cyclic stress as a surface load is directly applied to the section of the right end.

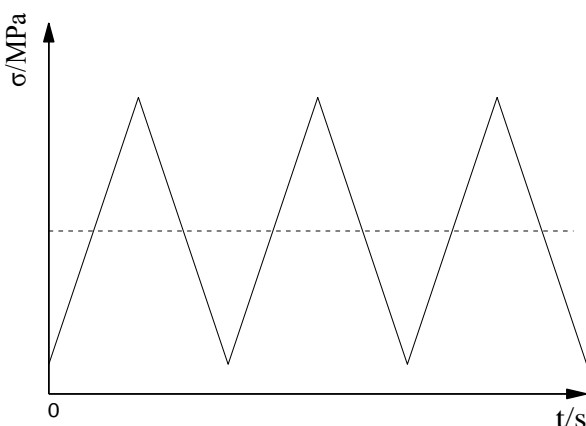

**Figure 8.** Schematic of loaded cyclic stress on the joint.

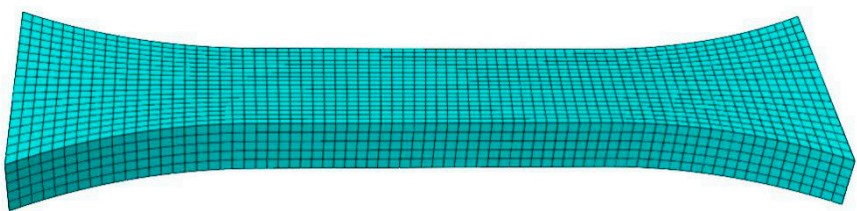

**Figure 9.** Meshed joint.

*4.3. Partitioned Performance Joint Model*

To compare the effectiveness of the continuous performance joint model for the fatigue performance simulation of the FSW joints, the partitioned performance joint model was established according to the metallurgical morphologies, hardness profile of the joints and the material attributes in different regions of the joint [25]. The joint is composed of the WNZ, TMAZ, HHAZ, LHAZ and BM, as shown in Figure 10. The fatigue parameters of different positions of the FSW joints were calculated according to the four-point correlation methods. The stress–strain data in different zones of the joint were obtained with Ramberg–Osgood equation. The material property does not have a change within each region.

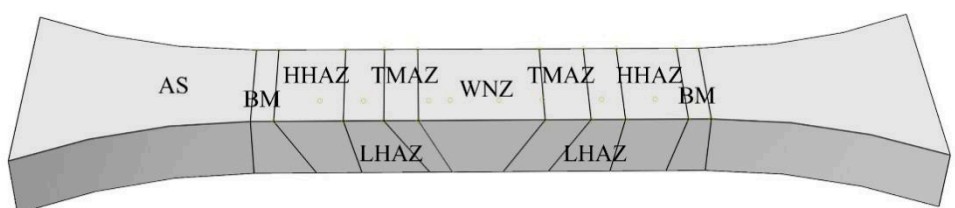

**Figure 10.** Partitioned performance joint model of the FSW joint.

## 5. Simulation Results

*5.1. Stress Distribution*

Von Mises stress distributions of continuous and partitioned performance joints are shown in Figure 11a,b. The stress distribution of the partitioned performance joint shows a significant mutation at the junction of different regions, especially at the junction of TMAZ and HAZ. The simulation results of continuous performance joints show the changes of stress and strain are smoother.

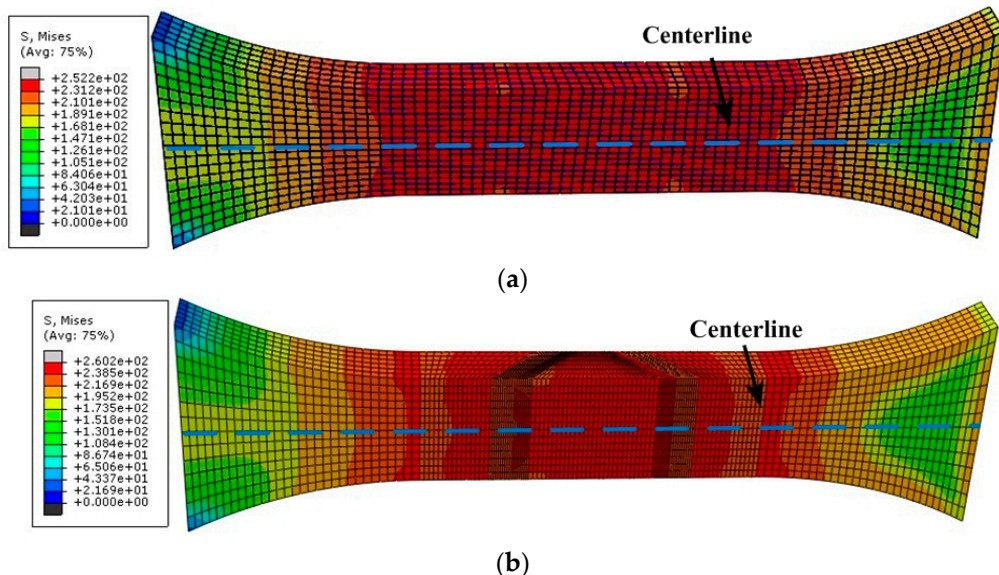

**Figure 11.** Von Mises stress contour plots of (**a**) continuous joint model and (**b**) partitioned joint model.

To observe the fatigue performance of the joint, the cyclic stress range of 216 MPa with the stress ratio of 0.1 was loaded on the joint. The maximum von Mises equivalent stress appeared in the upper surface of the joint in the partitioned performance joint model. The stress distribution trend of the upper and the lower surfaces in the continuous performance joint model is the same. The von Mises stress data of upper surface centerline were extracted from the simulation results of the continuous and the partitioned performance joint models. The maximum von Mises stresses both appeared in the HHAZ for the two models, as shown in Figure 12.

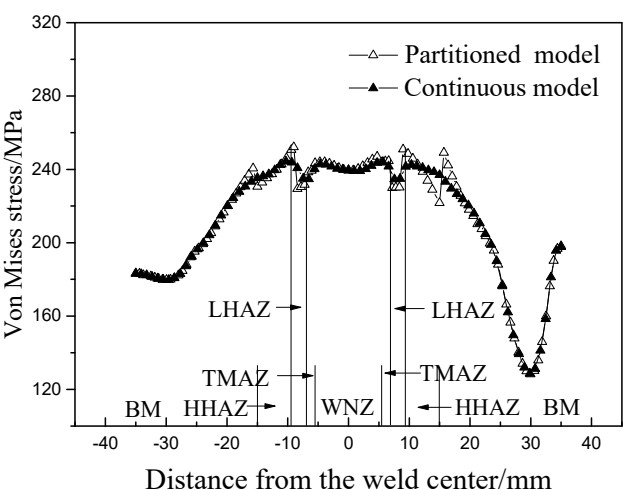

**Figure 12.** Von Mises stresses distributions of upper surface in the joint model.

The sudden change of stress in the partitioned performance joint model is obvious at the junction of adjacent regions in the joint. The distribution of von Mises equivalent stress obtained from the continuous performance joint model is relatively continuous for the joint, which is closer to the practical condition. The maximum stress appeared at the boundary of LHAZ and HHAZ for the partitioned performance joint model and it occurred at HHAZ near the LHAZ for the continuous performance joint model. The location of the maximum stress has a little difference for the two models.

The stress components of continuous and partitioned joint models were analyzed. There is no stress in the Z direction. The shear stresses are so small that their effects on the fatigue performance of the welded joint are ignored. It can be considered that the stresses in the X and the Y directions

determine the total stress distribution of the FSW joints. The stresses in the X and the Y directions of upper surface in the models were extracted for analysis as an example, as shown in Figure 13. The stresses in the X direction are larger in the two models since the tension-tension stress is loaded in the X direction. The stress fluctuation appeared in the X and the Y directions of the partitioned performance joint model. The stress distribution of the continuous performance joint model has no obvious sudden change.

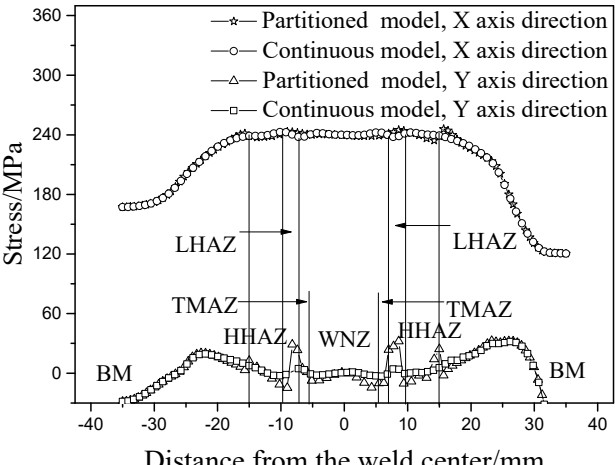

**Figure 13.** Stress distributions in the X and Y directions in upper surface of the joint model.

*5.2. Strain Distribution*

The weak area of joint fatigue performance can be determined by the stress and strain response of different zones in the joint from the simulation results of the fatigue property. The maximum principal strains at the centerline of upper surface, which extracted from the continuous and the partitioned performance joint models under the cyclic stress range of 216 MPa with a stress ratio of 0.1, are presented in Figure 14. Due to the abrupt change of material properties, the partitioned performance joint model caused a corresponding stress concentration at the junction of different regions and resulted in larger maximum principal strains than that of the continuous performance joint model. The larger the loaded cyclic stress in the model, the more obvious the abrupt change of strain distribution. The maximum principal strain is in the LHAZ near the TMAZ for the partitioned performance joint model and it appears in the HHAZ close to the LHAZ for the continuous performance joint model, which is approximately consistent with the fatigue failure position. The simulated fatigue weak area with the continuous performance joint model is more close to the fatigue experimental results.

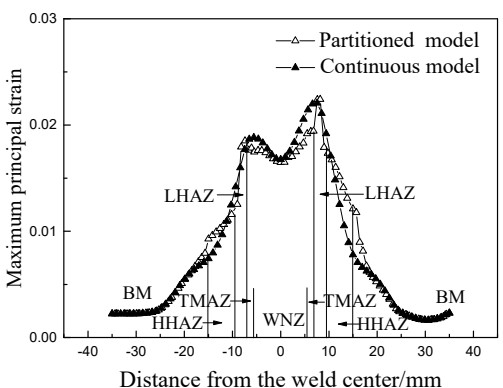

**Figure 14.** Maximum principal strain distributions of upper surface in the joint model.

By comparison with the partitioned performance joint model, the continuous performance joint model has a higher accuracy on predicting the fatigue weak areas of the FSW joint according to the

stress and strain distributions. It eliminates the large stress concentration caused by the abrupt change of material properties in the adjacent area and more accurately simulates the fatigue performance of the joint.

## 6. Fatigue Life Prediction

The joint models established in this paper not only evaluate the fatigue weak areas and obtain the stress and strain values of the joint, but they also predict the fatigue lives of the joints. In this section, the stress and strain values of the weak areas obtained from the continuous and the partitioned performance joint models are used to predict the fatigue lives of the FSW joints. The validity of the continuous performance joint model is further verified by comparison with the experimental results.

Crack initiation lives were estimated with Smith–Watson–Topper (SWT) method, which considered the effect of average stress [27]. The maximum principal stresses and corresponding maximum principal strain ranges of the weak area of the joint were extracted from the simulation results of the joint models. Fatigue parameters of weak areas are used to predict the fatigue crack initiation lives of joints. The SWT formula is shown below.

$$\sigma_{\max}\frac{\Delta\varepsilon}{2} = \frac{\sigma_f'^2}{E}(2N)^{2b} + \sigma_f'\varepsilon_f'(2N)^{b+c} \qquad (4)$$

where $\sigma_{\max}$ is the maximum principal stress, $\Delta\varepsilon$ is the maximum principal strain range, $N$ is the fatigue life.

The experimental results in Ref. [28] showed that the size of fatigue crack initiation was defined as 1 mm and the crack initiation lives for the HAZ and the WNZ of aluminum alloy welded joints were 40.21% and 60.67% of the total fatigue lives, respectively. The research in Ref. [29] showed that the crack initiation lives accounted for 40–50% of total fatigue lives. Therefore, the crack initiation life is selected as 50% of the total fatigue life in this paper.

The life prediction results of FSW joints are shown in Figure 15. The predicted errors of the fatigue lives based on the continuous and the partitioned performance joint models are basically within the factor of two by comparison with the experimental results. The fatigue life prediction results with the simulation data obtained from the continuous performance joint model are closer to the experimental lives because the continuous performance joint model has not the stress and strain concentrations caused by the area partition. It is shown that the continuous performance joint model is suitable for life prediction of the FSW joints.

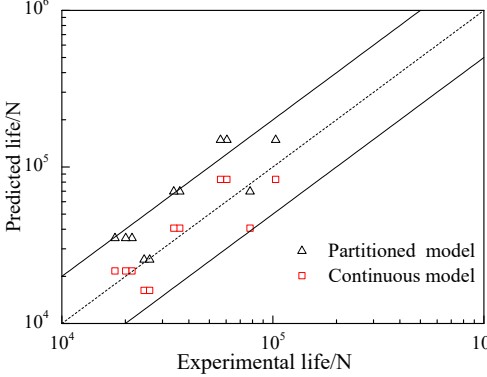

**Figure 15.** Life prediction of joint based on continuous and partitioned performance joint models.

## 7. Conclusions

(1) The original transverse and longitudinal residual stresses in the welded joint before fatigue are negative and the maximum tensile residual stress occurs in the HAZ during the tension–tension cyclic loading process.

(2) The fatigue parameters of different areas are obtained with four-point correlation method and the static mechanical property parameters of micro-tensile specimen at different locations of the joint. The continuous performance joint model is established by inputting different elastic moduli and fatigue stress–strain data at different locations of the FSW joint with user subroutine USDFLD. The stress components in the X and the Y directions determine the stress distribution of the FSW joints. The continuous performance joint model eliminates the stress and strain concentration caused by the area partition and more accurately simulates the fatigue performance of the joint by comparison with the partitioned performance joint model. The simulated fatigue weak area with the continuous performance joint model is more close to the fatigue experimental results.

(3) The fatigue life prediction of the FSW joints is proceeded with SWT method based on the simulation results of the continuous and the partitioned performance joint models. The results show that the predicted lives with the continuous performance joint model are closer to the experimental results and the life prediction error is within the factor of two.

**Author Contributions:** G.S. designed and performed the experiments; G.S., J.N. and X.W. analyzed the experimental data and simulated the joint model; D.S. and S.C. gave some advices for the research; G.S. and X.W. wrote the paper.

**Funding:** This research was founded by the National Natural Science Foundation of China (Grant No. 11672010, 51535001 and 51575012).

**Conflicts of Interest:** The authors declare no conflicst of interest.

## Abbreviations

| | |
|---|---|
| FSW | Friction stir welding |
| WNZ | Weld nugget zone |
| TMAZ | Thermo-mechanically affected zone |
| HAZ | Heat affect zone |
| BM | Base material |
| HHAZ | High-hardness heat affect zone |
| LHAZ | Low-hardness heat affect zone |
| AS | Advancing side |
| RS | Retreating side |
| SWT | Smith-Watson-Topper |

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
