# Peer review of "Influence of Residual Stress on Fatigue Weak Areas and Simulation Analysis on Fatigue Properties Based on Continuous Performance of FSW Joints"

_metals, doi:10.3390/met9030284_

Reviewer 1 Report

The following comments are given by the reviewer (answers/discussions should be preferably included in the revised version of the paper):

- Abstract: What is the meaning of “fatigue simulation”? This is not a proper definition. Quantitative results should be added to highlight the applicability of the presented approach.

- Chapter 2.1: Nominal mechanical properties (yield strength, etc.) of the base material should be added. How were the specimens exactly manufactured? Within an assembly with post machining to the final specimen geometry? Add more detailed information.

- Figure 2: Where these zones are exactly located (at the cross section of one specimen)? What is the dimension of each zone? An additional figure illustrating an overview of the whole microstructure within the welded zone would be beneficial.

- Figure 3: It is not common to show the maximum stress, please use stress range instead. What was the failure criterion of the tests (total fracture, crack initiation)? How was the statistical analysis conducted (which procedure, reference)? Add parameters of the S/N-curve (slope, scatter band, etc.)

- Chapter 2.4: How were the residual stress measurements conducted? (collimator size, exposure time, etc.) Add more details.

- Line 111 to 114: The authors state that original residual stresses do not have an effect on fatigue. Why? In general, residual stresses can have a major impact on fatigue.

- Figure 5 and 6: The difference between original and fatigue damaged needs to be explained scientifically. Just showing differences is not a scientific explanation.

- Chapter 3: The simulation is not well described at all. Which fatigue model was exactly used? (equations, references)? How were the parameters exactly determined (experimental results, illustration of parameter evaluation, etc.)? Why are the given differences for the parameters shown in Table 2 (scientific explanation)? On which basis are Equ 1 to 4 defined (references)?

- Figure 9: The authors only consider the different material properties for the different microstructures. However, residual stresses are totally neglected, which may lead to completely wrong results – the authors must prove that this procedure is applicable.

- Figure 10: von Mises stresses are shown. What is the origin of these stresses? (further information needs to be provided)

- Conclusions: Based on the comments, the current conclusions may not be scientifically proven due to some missing information for the experimental and numerical work. Please add accordingly. Additionally , the conclusions should be supported by quantitative results, as commented for the abstract.

Author Response

Dear Reviewer #1,

We are very grateful to your valuable comments. We have carefully taken the comments into consideration in preparing our revision and tried our best to revise the manuscript according to your advice. The point-to-point responses to your comments are listed below.

1. Abstract: What is the meaning of “fatigue simulation”? This is not a proper definition. Quantitative results should be added to highlight the applicability of the presented approach.

Response: “simulation analysis of fatigue property” is the exact meaning. We have revised the words of “fatigue simulation”. The contents of life prediction are added to highlight the applicability of the presented continuous performance method.

2.   Chapter 2.1: Nominal mechanical properties (yield strength, etc.) of the base material should be added. How were the specimens exactly manufactured? Within an assembly with post machining to the final specimen geometry? Add more detailed information.

Response: The nominal mechanical properties of the base material, the manufactured process of the fatigue specimens and some other detailed information have been added in the manuscript.

3.  Figure 2: Where these zones are exactly located (at the cross section of one specimen)? What is the dimension of each zone? An additional figure illustrating an overview of the whole microstructure within the welded zone would be beneficial.

Response: The cross-sectional metallographic morphology of one specimen has been added in the manuscript. The dimension of each zone can be obtained from the hardness profile, in which all regions are identified. Figure 2 in the revised manuscript is an overview of metallographic morphology of the joint.

4.  Figure 3: It is not common to show the maximum stress, please use stress range instead. What was the failure criterion of the tests (total fracture, crack initiation)? How was the statistical analysis conducted (which procedure, reference)? Add parameters of the S/N-curve (slope, scatter band, etc.)

Response: The expression of maximum stress has been replaced by cyclic stress range. The failure criterion of the test is based on the fracture of the specimen, which has been added in the manuscript. The fitted equation of the S-N curve is added in Figure 3.

5. Chapter 2.4: How were the residual stress measurements conducted? (collimator size, exposure time, etc.) Add more details.

Response: Sorry, the residual stress measurements were finished in the centre for instrumental analysis of our university. The measurements operation was conducted by the lab master of that center. We just got the stress results without the parameters of the measurements.

6.  Line 111 to 114: The authors state that original residual stresses do not have an effect on fatigue. Why? In general, residual stresses can have a major impact on fatigue.

Response: The whole distribution of original residual stress in the joint that measured with GADDS X-ray diffraction meter is basically negative or near zero. The negative stress is beneficial to the fatigue life. In our research, the loading cyclic stress is positive tension-tension load. Thus, the original residual stress will be relaxed soon. The surface stress will change to positive once the cyclic tensile stress is loaded. Therefore, we didn’t consider the original residual stress in the numerical simulation.

7.   Figure 5 and 6: The difference between original and fatigue damaged needs to be explained scientifically. Just showing differences is not a scientific explanation.

Response: Some explanations have been added for the difference between the original residual stresses and the residual stresses of fatigue-damaged specimens.

8.   Chapter 3: The simulation is not well described at all. Which fatigue model was exactly used? (equations, references)? How were the parameters exactly determined (experimental results, illustration of parameter evaluation, etc.)? Why are the given differences for the parameters shown in Table 2 (scientific explanation)? On which basis are Equ 1 to 4 defined (references)?

Response:

Which fatigue model was exactly used?

The Ramberg-Osgood equation is used to calculate the cyclic stress and strain data that are needed to input in the material attribute. The stress and strain responses in the numerical simulation are the cyclic stress and strain in the steady state under cyclic loading since the constitutive relation is cyclic stress–strain and the cyclic stress is loaded.

On which basis are Equ 1 to 4 defined?

The original empirical equations 1 to 4 are algebraic manipulation with four-point correlation method [28]. The illustration of four-point correlation method has been added in the manuscript. 

How were the parameters exactly determined?

The fatigue parameters can be calculated with the static mechanical properties of the micro-tension specimens in different zones using the four- point correlation method. They will be used in the cyclic strength coefficient and exponent that needed for Ramberg-Osgood equation and the life prediction equation.

Why are the given differences for the parameters shown in Table 2?

The fatigue parameters are different in different zones because the materials of different zones in the joint have different microstructures and different mechanical properties.

9. Figure 9: The authors only consider the different material properties for the different microstructures. However, residual stresses are totally neglected, which may lead to completely wrong results – the authors must prove that this procedure is applicable.

Response: Because the measured original residual stress is not high and negative and it will be relaxed soon once the tension-tension cyclic stress is loaded. Therefore, it did not considered to add on the joint as a prestress in the fatigue property numerical analysis under the tension-tension cyclic loading.

10. Figure 10: von Mises stresses are shown. What is the origin of these stresses? (further information needs to be provided)

Response: The description of stress components of the X and the Y directions has been added in the Section 5.1: Stress distribution. 

11. Conclusions: Based on the comments, the current conclusions may not be scientifically proven due to some missing information for the experimental and numerical work. Please add accordingly. Additionally, the conclusions should be supported by quantitative results, as commented for the abstract.

Response: Some explanations and contents have been added in the paper and the conclusion has been revised.

Reviewer 2 Report

The manuscript describes the distribution of residual stresses in FSW joints and their influence on fatigue weak area and a simulation of fatigue behaviour of FSW joints.

The study is interesting, but not very clearly presented, so it could not be accepted in its actual version. It is recommended that the following comments are addressed and answered by the authors:

           Introduction section: it is too synthetic. Please deepen the state of the art and highlight the novelty of your work compared to the existing ones; the application of continuous performance model and partitioned performance model on welded joints must be discussed and the main difference in terms of expected response must be added;   

-          In the experimental section a clearer description of the welding process could be inserted. For example, did the welding parameters derive from previous studies?

-       Please clearly separate the description of investigation methods from the results. In the same way also the simulation models have to be described separately from the results obtained;

-          In Section 2.2, how do the authors understand that fracture occurs in HHAZ and not generically in HAZ? Please discuss that.

-        In Section 2.3 authors said: “The minimum hardness values of the upper and lower surfaces are both in the LHAZ according to the hardness distribution of the joints, which is not in coincidence with the failure position of HHAZ for most fatigue specimens. The results show that the minimum hardness location of the joints may not be necessarily the fatigue weak areas of joints.” Please give a scientific explanation of these results.

-          How are the experimental and the simulation parts of the studies linked? From their current description, they seem completely disconnected.

-          No details about the residual-stress investigation method is given;

-          In the discussion of the results, scientific explanation and comparison with references and previous studies on same topics are totally missing.

 Some minor modifications:

-          Table 1 and Figure 7 captions: please uniform the style…some letters are bigger than others.

-          Lines 100-101: correct reference is to figure 5a and 5b and not 4 and 5.

-          Figure 5: please scale the graphs in the same way in order to give a better comparison of the results.

-          Figure 11: are the graphs referring to lower surface centerline? If yes, please add this information in the caption.

-          Line 223: please change “stain” to “strain”.

Author Response

Dear reviewer #2,

We are very grateful to your valuable comments for the manuscript. We tried our best to revise the manuscript according to your advice. The point to point responses to your comments are listed below.

 1.  Introduction section: it is too synthetic. Please deepen the state of the art and highlight the novelty of your work compared to the existing ones; the application of continuous performance model and partitioned performance model on welded joints must be discussed and the main difference in terms of expected response must be added;   

ResponseWe have added some contents to highlight the novelty of our work in the introduction section. The part of life prediction has been added for further exhibiting the application and the difference of the two joint models except the stress analysis. Some other differences in the establishing process have been explained in the Section 4: Fatigue numerical simulation.

2. In the experimental section a clearer description of the welding process could be inserted. For example, did the welding parameters derive from previous studies?

ResponseSome descriptions of the welding process have been added in the paper. The welding was finished in the China Academy of Launch Vehicle Technology. The technologists of this academy provided the welding parameters and the high-quality joints to us.

3. Please clearly separate the description of investigation methods from the results. In the same way also the simulation models have to be described separately from the results obtained;

ResponseThe experimental and simulation results have been separated from the experiment and the simulation models.

4. In Section 2.2, how do the authors understand that fracture occurs in HHAZ and not generically in HAZ? Please discuss that.

ResponseThe HAZ can be further divided into high-hardness heat affect zone (HHAZ) and low-hardness heat affect zone (LHAZ) based on the hardness distribution, the grain sizes and the metallographic morphologies. The phase in the HHAZ has the similar size as that in the BM, while the hardness in HHAZ is lower than that of BM. The fracture locations of the specimens can be judged by combining the coordinates of the fracture locations of the specimens, the hardness profiles and the metallographic morphologies.

5. In Section 2.3 authors said: “The minimum hardness values of the upper and lower surfaces are both in the LHAZ according to the hardness distribution of the joints, which is not in coincidence with the failure position of HHAZ for most fatigue specimens. The results show that the minimum hardness location of the joints may not be necessarily the fatigue weak areas of joints.” Please give a scientific explanation of these results.

ResponseSome explanations have been added in the paper.

The hardness has a relation with the material strength but the fatigue weak area of the joint is not necessarily corresponding to the location of minimum hardness [27]. We think the fatigue weak area has a relation with the variation of the hardness gradient, which corresponds to the variation of the mechanical property and heterogeneous microstructure.

6.   How are the experimental and the simulation parts of the studies linked? From their current description, they seem completely disconnected.

ResponseThe fatigue experimental results are used for the verification of the simulation results. The residual stress measurements suggest that the original residual stress of the joint specimen is negative and not great. It will be relaxed once the tension-tension cyclic stress is loaded. So we ignored the original stress and did not load the original residual stress on the joint as a prestress. The explanation has been added in the paper.

7. No details about the residual-stress investigation method is given;

ResponseSorry, the residual stress measurements were finished in the instrumental analysis center of our university. The measurements operation was conducted by the lab master of that center. We just got the stress results without the parameters of the measurements.

8.  In the discussion of the results, scientific explanation and comparison with references and previous studies on same topics are totally missing.

ResponseSome explanations and comparison with references have been added in the simulation process and results.

9. Some minor modifications:

 Table 1 and Figure 7 captions: please uniform the style…some letters are bigger than others.

 Lines 100-101: correct reference is to figure 5a and 5b and not 4 and 5.

Figure 5: please scale the graphs in the same way in order to give a better comparison of the results.

  Figure 11: are the graphs referring to lower surface centerline? If yes, please add this information in the caption.

  Line 223: please change “stain” to “strain”.

Response

The style has been uniformed for all the tables and figures.

The graph labels for all the figures have been examined and corrected.

We have marked the centreline in Figure 11.

Other mistakes have also been corrected.

Round  2

Reviewer 1 Report

The revised paper is improved according to the comments by the reviewer, which is fine.

Reviewer 2 Report

The paper significantly improved and could be accepted in its present form.